# Investigation of gate dielectric interface on contact resistance of short channel organic thin film transistors (OTFT)

**Lingala Prasanthi[1], Asisa Kumar Panigrahy[2], Subhashini Tata[3], Rohith Bala Jaswanth B.[4], Talla Srinivasa Rao[5], Matta Durga Prakash**📶[1]*, **Shovan Kumar Kundu**📶[6]

**1** Department of ECE, SRM University-AP, Mangalagiri, Andhra Pradesh, India, **2** Department of ECE, Faculty of Science and Technology (IcfaiTech), ICFAI Foundation for Higher Education Hyderabad, Hyderabad, Telangana, India, **3** Department of ECE, Seshadri Rao Gudlavalleru Engineering College, Gudlavalleru, Andhra Pradesh, India, **4** Department of Internet of Things, Seshadri Rao Gudlavalleru Engineering College, Gudlavalleru, Andhra Pradesh, India, **5** Department of ECE, Aditya University, Surampalem, Andhra Pradesh, India, **6** Department of Physics, Faculty of Science and Technology (FST), American International University-Bangladesh, Bangladesh

* durgaprakash.m@srmap.edu.in

## Abstract

This study provides a comprehensive analysis of the impact of the interfacial properties on the performance of organic thin-film transistors (OTFTs) with a hybrid dielectric of $Al_2O_3$/PVP compared to single-layer dielectrics of $Al_2O_3$ and PVP. The analyses were performed using the 2D Silvaco Atlas numerical simulator, which conducted a detailed numerical investigation into how varying the thickness ratio of $Al_2O_3$ and PVP in the dielectric affects contact resistance and off-state current in short-channel OTFTs. High-K dielectric materials, such as $Al_2O_3$, offer low threshold voltages but lead to increased contact resistance and leakage current, while low-K dielectrics like PVP reduce leakage current but suffer from lower mobility and higher contact resistance. By utilizing a hybrid $Al_2O_3$/PVP dielectric, we successfully reduced the contact resistance to 4.84 KΩ.cm², as extracted from $V_{DS}$-$I_D$ characteristics at a gate voltage of -2V. Additionally, contact resistance significantly influenced the off-state current, particularly in devices of short channel length (1 μm). The PVP layer, with thicknesses ranging from 2.4 nm to 4.2 nm, effectively reduced charge carrier traps at the semiconductor/dielectric interface, enhancing mobility. Furthermore, hysteresis effects were examined through C-V characteristics by sweeping the gate voltage from -3V to +3V. These findings highlight the trade-offs in optimizing PVP thickness to balance interface quality and electrical performance in hybrid dielectric OTFTs.

## Introduction

Over the last few decades, organic semiconductors have been critical to advancing organic thin-film transistor technology. These transistors demonstrate a significant

**Data availability statement:** All relevant data are available at the following: lingala, prasanthi; Prakash, Dr. M Durga; Panigrahy, Asisa Kumar; Kumar Kunda, Shovan; Tata, Subhashini; Bala Jaswanth B, Rohith; et al. (2025). Investigation of Gate Dielectric Interface on Contact Resistance of Short Channel Organic Thin Film Transistors (OTFT). Figshare. Dataset. https://doi.org/10.6084/m9.figshare.29608790.v1.

**Funding:** The author(s) received no specific funding for this work.

**Competing interests:** The authors have declared that no competing interests exist.

advancement in the field of flexible and wearable electronics due to their distinctive characteristics, such as flexibility, low-temperature processing at fabrication, and being suitable to fabricate on plastic substrates [1]. They serve as controller circuits for numerous applications, including active-matrix organic light-emitting diodes (AMOLEDS) [2], RFID tags [3], memory applications [4], gas and chemical sensors, biosensors [5,6], large flat panel displays [7], and so on. Extensive studies have been carried out on different organic semiconductors and dielectric layers crucial for improving OTFT performance. To achieve higher switching speeds and operate at high frequencies, scaling down the channel length is required. However, this approach will be ineffective unless contact resistance is minimized, as it adversely affects the performance of short-channel OTFT devices. Consequently, extensive research has been conducted to minimize contact resistance through methods such as contact doping [8,9], using compatible contact materials, and employing self-assembled monolayer (SAM) techniques [10,11]. Despite these efforts, several factors affect contact resistance, including the electron affinity of organic semiconductors, interface traps at the dielectric/semiconductor junction, and interface dipoles. The interface between the gate dielectric and the semiconductor significantly influences the electrical performance of an OTFT, affecting parameters like threshold voltage, mobility, and charge carrier accumulation at low gate voltages [12,13].

A number of researchers have explored how high-K inorganic dielectrics like $Al_2O_3$ [14], $HfO_2$ [15], $Ta_2O_5$ [16], $ZrO_2$ [17], and $TiO_2$ [18] might be able to improve the performance of OTFTs. High-K materials possess a high dielectric constant, so increasing the dielectric gate capacitance can accumulate more charge carriers in the dielectric-semiconductor interface when a low $V_{GS}$ voltage is applied. Consequently, the operating voltage is reduced. The issue is that the majority of high-K materials possess a narrow band gap, resulting in a leakage current between semiconductors and dielectric interface. FC chiu, [19] stated that the chosen dielectric should have a high dielectric constant, a wide band gap, minimal interface traps, and great thermal stability. However, the high K dielectric constant causes a high surface energy, which lets organic semiconductors deposit unevenly on the dielectric surface. This compromises the interface's quality, leading to defects and an increase in interface trap density. These consequences include a decline in mobility and enhances the contact resistance [20], as well as the need a high temperature for processing the high-K materials.

In order to address this issue, organic dielectric is deposited over high-k materials. This renders the dielectric surface hydrophobic, making it suitable for depositing organic semiconductors, results an organic-organic interface with a noninteracting interface [21,22]. This enhances interface quality and device performance. Organic dielectrics like polyvinyl phenol (PVP), polyvinyl alcohol (PVA), polyimide (PI), polymethylmethacrylate (PMMA), and others that have been proposed as gate dielectrics for organic electronics typically exhibit low dielectric constants, resulting in low capacitance, reduced mobility, high operating voltages, and hysteresis loss. A solution was proposed to utilize bilayer gate dielectrics made up of high-k metal-oxide films covered with low-k polymer films. This would combine the benefits of

high-k oxide, which has a high capacitance and low operating voltages, with polymer dielectrics, which have better inter-faces with organic semiconductors. Numerous researchers have developed the bilayer dielectric structure of OTFT in order to incorporate the benefits of low-K organic polymers to preserve a good interface quality with the organic semicon-ductor and high-K dielectrics for high capacitance. Liwei Shang and Ming Liu et al. [21] fabricated an OTFT by combining $ZrO_2$ and PMMA as a bilayer dielectric with CuPu as the organic semiconductor. This design showed reduced leakage current compared to using only a single layer of $ZrO_2$. Sagarika Khound et al. [23] examined the electrical properties of a hybrid dielectric made of pentacene-based cross-linked polyvinyl phenol (cPVP) and lanthanum oxide ($La_2O_3$) using a solution processing approach.

Still, all these bilayer devices were operating at voltages higher than 15 V. Hence, it is essential to develop an OTFT that operates at low voltages and is appropriate for portable uses such as sensors and wearable electronics. Developing OTFTs that can operate effectively at low voltage and high switching speed without compromising electrical performance or operational stability is still a significant problem. On the other hand, investigations on contact resistance and leakage current in optimizing the hybrid dielectric layer have been rarely reported. There are only a few studies related to exploring the impact of dielectric on contact resistance [24–26]. Still, all these studies utilized single gate dielectric layers and there-fore may not completely eliminate the influence of interface charge trap density and high surface energy at the interface.

In this context, we chose $Al_2O_3$/PVP as the hybrid dielectric stack, due to their complementary material properties and proven compatibility with organic semiconductors. $Al_2O_3$ offers a high dielectric constant (~9.3), wideband gap, excellent thermal stability, and favorable electrical performance at low voltages. Meanwhile, PVP contributes hydrophobicity, low interface trap density, mechanical flexibility, and biocompatibility. This specific combination was selected over other stacks because (i) $Al_2O_3$ provides better interfacial quality and lower trap density compared to high -K dielectric materials which often introduces instability in OTFTs, and (ii) PVP exhibits better uniform film formation and fewer residual polar groups than organic dielectrics, which improves the semiconductor–dielectric interface. These advantages make $Al_2O_3$/PVP a robust choice for hybrid dielectric engineering in high performance OTFTs.

This paper introduces a novel bottom gate OTFT based on dinaphtho[2,3 b:2',3'-f]thieno[3,2-b]thiophene (DNTT) as an organic semiconductor and employing a hybrid gate dielectric $Al_2O_3$/PVP, which minimizes the compromises between operating voltage and contact resistance in the short channel OTFTs as shown in Fig 1. Here, we explored the effect of single and hybrid dielectric layers of $Al_2O_3$ and PVP on the device parameters of the OTFT. We specifically investigated the impact of different interfacial effects on the electrical characterization of OTFTs, focusing on single dielectrics as well as various PVP thickness compositions within the hybrid dielectric layer. Although the devices are structurally identical, they differ in the composition of the PVP/$Al_2O_3$ hybrid dielectric layer. This variation allows us to understand the impor-tance of optimizing PVP thickness in the hybrid dielectric composition and its impact on device performance. In this work, the simulated devices will be named according to the following nomenclature: "P##A##," where "##" refers to the ratio of

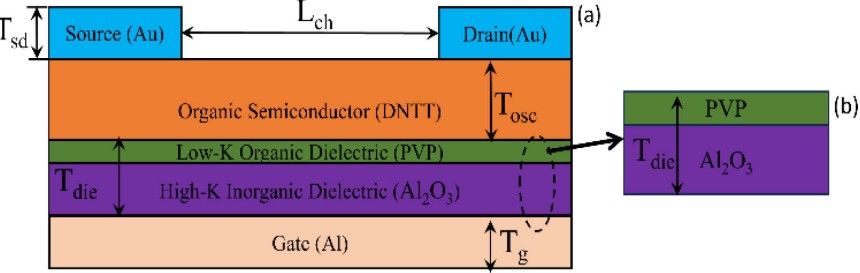

**Fig 1. (a) A schematic representation of the simulated device structure for DNTT OTFTs (b) Varying dielectric layer single $Al_2O_3$, single PVP and hybrid layer with different thickness composition (P5A95, P20A80 and P40A60).**

thickness of the dielectric layer measured in percentage, "P" refers to the PVP layer, and "A" to the $Al_2O_3$ layer. Following this nomenclature, a device named P5A95 comprises a 5% PVP layer (0.6 nm) and a 95% $Al_2O_3$ layer (11.4 nm) out of 12 nm, P20A80, and P40A60. The P20A80 OTFT exhibited lower contact resistance compared to the other devices. It was observed that contact resistance affects both the off-state current and the mobility of the devices. We further examined hysteresis loss using CV characteristics by sweeping the gate voltage from -3V to +3V. The P20A80 bilayer dielectric layer device operates at 3 V, features a high $I_{ON}/I_{OFF}$ ratio of $1.83 \times 10^{10}$, a subthreshold swing of 57 mV/dec, low OFF- state current and negligible hysteresis loss.

## Device modelling and simulation

### Modelling of parameters

The $I_{DS}$-$V_{GS}$ characteristics of the OTFT are employed to extract the device parameters, which include mobility, transconductance, sub-threshold voltage, $I_{ON}/I_{OFF}$ ratio, and threshold voltage. The $I_{DS}$-$V_{GS}$ features are separated into two distinct regions: a linear region and a saturation region. Within the linear region, the accumulation of charge carriers occurs at the interface between the dielectric and semiconductor when $V_{DS}$ is less than ($V_{GS}$-$V_{TH}$). In this case, the drain current of an OTFT based on DNTT can be determined in Equation (1) [27].

$$I_D = \frac{W}{L}\mu C_{ox}(V_{GS} - V_{TH})V_{DS} \tag{1}$$

Where W is width, L is length $C_{ox}$ represents oxide capacitance and µ is mobility.

For hybrid dielectric configurations ($Al_2O_3$/PVP), $C_{ox}$ was calculated as the effective series capacitance of the two dielectric layers,

$$\frac{1}{C_{OX}} = \frac{1}{C_{Al2O3}} + \frac{1}{C_{PVP}} \tag{2}$$

where $C_{Al2O3}$ and $C_{PVP}$ are the individual capacitances derived from their respective dielectric constants and physical thicknesses. For single-layer devices, $C_{ox}$ was computed using the standard parallel-plate formula.

When $V_{DS}$ exceeds ($V_{GS}$-$V_{TH}$) in the saturation area, $I_{Dsat}$ is given by Equation (3)

$$I_{Dsat} = \frac{W}{2L}\mu C_{ox}(V_{GS} - V_{TH})^2 \tag{3}$$

$$\sqrt{I_{Dsat}} = \sqrt{K}(V_{GS} - V_{TH}) \tag{4}$$

Equation (3) can be expressed as Equation (4), with $\sqrt{K}$ representing the slope and $V_{TH}$ denoting the threshold voltage. To extract the threshold voltage of an organic thin-film transistor (OTFT), perform a linear fit on the x-intercept slope of the square root of the drain current $\sqrt{I_{Dsat}}$ versus the gate-source voltage $V_{GS}$ plot. This linear fit can then be extrapolated on the x-axis to obtain the desired threshold voltage value. Transconductance ($g_m$), a critical parameter for a transistor, is obtained by taking the derivative of Eq. (3) with respect to $V_{GS}$.

It evaluates the better switching capacity and operational efficiency of the device, which can be expressed as Equation (5).

$$g_m = \frac{\partial I_{DS}}{\partial V_{GS}} \tag{5}$$

The field effect mobility Equation (6) can be determined by calculating the transconductance ($g_m$) provided by,

$$\mu_{eff} = \frac{Lg_m}{wC_{ox}} \times \frac{1}{V_{DS}}$$

(6)

Thus, the carrier mobility in the saturation region can be computed from Equation (4) as,

$$\mu_{sat} = \frac{2L}{WC_{ox}} \left( \frac{d\sqrt{I_{DS}}}{dV_{GS}} \right)^2$$

(7)

## Structural design

The schematic of the proposed 2D dimensional bottom gate top contact DNTT-based OTFT device with the $Al_2O_3$/PVP bilayer dielectric structure with dimensions of 1 μm channel length (L) and 200 μm width (W) utilized in this simulation is illustrated in Fig 2(a). The aluminium with a thickness of 30 nm can act as a gate as well as a substrate for the device. Next, deposit high-K $Al_2O_3$ over the gate electrode, and organic PVP can be covered over the $Al_2O_3$ to provide a suitable surface for uniform deposition of the organic semiconductor. The total thickness of the hybrid dielectric layer is 12nm. Then an organic semiconductor, DNTT, is deposited over the top of the PVP layer with a thickness of 25 nm, thus providing an organic-organic interface. Finally, deposit gold material of 30 nm as a source and drain contact on the layer of organic semiconductor. The structural and material parameters are derived from Kraft et al. [28] and summarized in Table 1.

The performance of the DNTT-based OTFT is investigated by varying the single and hybrid dielectric layers of $Al_2O_3$ and PVP with the help of Silvaco atlas device simulator. The overall thickness of dielectric layer maintains constant throughout the simulations but varied the ratio of thickness composition of $Al_2O_3$ and PVP in the hybrid dielectric structures, that namely as P5A95, P20A80, and P40A60 as shown in Fig 2. Although such ultra-thin polymer layers are challenging to fabricate, they are achievable using controlled spin-coating techniques, where the film thickness is modulated through solution concentration and spin speed. Fan et al. [29] demonstrated the successful deposition of sub-5 nm PVP layers atop a PVA layer, which improved organic thin-film transistor performance by enhancing surface morphology and enabling better molecular ordering of the semiconductor layer. Additionally, post-deposition thermal cross-linking improves film uniformity and mechanical stability [28]. These findings affirm the experimental feasibility of integrating ultra-thin PVP interfacial layers in high-performance OTFT devices, consistent with our simulation assumptions.

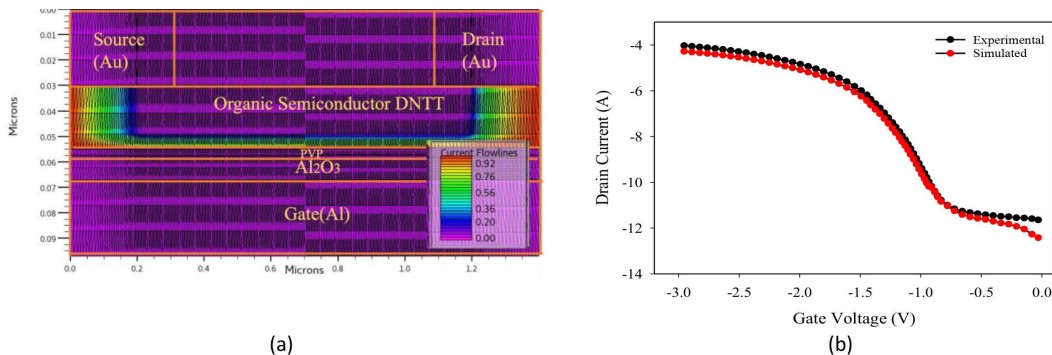

(a)                                                                        (b)

**Fig 2. (a) Simulated structure of top contact bilayer gate dielectric OTFT representing current flow lines at conducting channel (b) Comparison of experimental and simulated transfer characteristics of OTFT at 300K [28].**

**Table 1. Summary of structure dimensions and material parameters used in simulation.**

| Classification | Parameters | Values/units |
|---|---|---|
| Physical Dimensions | Channel Length ($L_{ch}$) | 1 µm |
| | Width (W) | 200 µm |
| | Thickness of DNTT ($T_{osc}$) | 25 nm |
| | Thickness of single dielectric $Al_2O_3$, PVP ($T_{die}$) | Fixed at 12 nm |
| | Thickness of hybrid dielectric $Al_2O_3$/ PVP ($T_{die}$) | Varied for different compositions |
| | Thickness of electrodes($T_{sd}$,$T_g$) | 30 nm |
| Materials Organic SC-D NTT | Bandgap ($E_g$ 300) | 3.38 [34] |
| | Permittivity (ε) | 3 |
| | Electron Affinity ($\chi_e$) | 1.81 |
| | Effective DOS of LUMO ($N_c$) | 1e20 cm$^{-3}$ |
| | Effective DOS of HOMO($N_v$) | 1e20 cm$^{-3}$ |
| | Doping Concentration | 1x10$^{15}$ cm$^{-3}$ |
| $Al_2O_3$ | Dielectric Constant | 9.3 [35] |
| | Bulk trap density | 1x10$^{14}$cm$^{-3}$ |
| PVP | Dielectric Constant | 3.8 |
| Work function | Gold (Au) | 5.0 eV |
| | Aluminium (Al) | 4.28 eV |
| PF Mobility Parameters | deltaep.pfmob | 1.792e-1 eV |
| | betap.pfmob | 7.785e5 eV(cm/V)$^{1/2}$ |
| | Temperature (K) | 300K |

**Simulation environment and numerical solver.** The simulations were performed using Silvaco Atlas, a TCAD tool that analyzes both DC and transient characteristics by solving the self-consistent Poisson and drift-diffusion equations. This tool predicts and optimizes the performance of electronic devices and circuits prior to fabrication. The device structure was discretized using a triangular mesh, optimized for precision and computational efficiency as shown in Fig 2(a). Boundary conditions were assigned as, the source and drain contacts were treated as Schottky-type boundaries with defined work-functions, the gate contact was modeled as an ideal metal with Dirichlet potential. All non-contact edges were treated as Neumann (zero-flux) boundaries. Interface continuity and trap behavior were incorporated using the INTERFACE and INTDEFECTS models.

To model the DNTT layer, the simulation incorporates the Langevin recombination concept, Shockley-Read-Hall (SRH) processes, trap states, as well as Poole-Frenkel and hopping mobility. The PF mobility model establishes the relationship between mobility capabilities and the interaction with the electrical field. This model is mathematically represented as [30,31].

$$\mu(E) = \mu(0)exp\left(\frac{\beta\sqrt{E}}{KT}\right)$$

(8)

Where the zero field mobility,

$$\mu(0) = \mu_i exp\left(-\frac{\Delta}{KT}\right)$$

(9)

where μ(E) is the field dependent mobility, β is the Poole–Frenkel factor, E is the electric field, $\Delta$ is the zero field thermal activation energy, and μ$_i$ is the intrinsic mobility. Table 1 enumerates the material input parameters, including device dimensions, dielectric properties, and mobility models used in the TCAD simulations for OTFTs with hybrid and single-layer dielectrics.

To account for the effect of trapped charges, Poisson's equation was modified to incorporate spatial charge contributions from ionized traps, expressed as Eq. (10).

$$div(\varepsilon \nabla \Psi) = q(n - p - N_{tD}^+ + N_A^-) - Q_T \tag{10}$$

where $Q_T = q\left(N_{tD}^+ + N_{tA}^-\right)$ represents the total trapped charge density

Here, $N_{tD}^+ = trap\ density\ \times\ F_{tD}$ and $N_{tA}^- = trap\ density\ \times F_{tA}$

Here, $N_{tD}^+$ and $N_{tA}^-$ are the ionized donor- and acceptor-like trap densities respectively and $F_{tD}$ and $F_{tA}$ are their respective ionization probabilities. Organic semiconductors inherently, contain a tail and deep -level traps within the band gap, which are described using both exponential (tail) and Gaussian (deep) density-of-states (DOS) functions.

The equations that describe these terms are as follows,

$$g(E) = g_{TA}(E) + g_{TD}(E) + g_{GA}(E) + g_{GD}(E) \tag{11}$$

where the tail and deep level bands are modeled with an exponential and Gaussian distributions, respectively, as:

$$g_{TA}(E) = N_{TA} exp\left[\frac{E - E_c}{W_{TA}}\right] \tag{12}$$

$$g_{TD}(E) = N_{TD} exp\left[\frac{E_V - E}{W_{TD}}\right] \tag{13}$$

$$g_{GA}(E) = N_{GA} exp\left[-\left[\frac{E_{GA} - E}{W_{GA}}\right]^2\right] \tag{14}$$

$$g_{GD}(E) = N_{GD} exp\left[-\left[\frac{E - E_{GD}}{W_{GD}}\right]^2\right] \tag{15}$$

Here, E is the trap energy, E$_c$ is the conduction band energy, E$_V$ is the valence band energy and the subscripts (T, G, A, D) stand for tail, Gaussian (deep level), acceptor and donor states respectively. The quantities N$_{TA}$, N$_{TD}$, N$_{GA}$, N$_{GD}$ and W$_{TA}$, W$_{TD}$, W$_{GA}$, W$_{GD}$ are the fitting parameters of TRAP models.

In our simulations, both interface and bulk trap effects were incorporated using Silvaco Atlas INTDEFECTS and INTERFACE models to accurately capture semiconductor-dielectric and semiconductor-contact interfacial behaviors. Specifically, we implemented a Gaussian distribution of bulk traps in the Al$_2$O$_3$ layer with a peak trap density of $1 \times 10^{14}$ cm$^{-3}$, centered at 0.6 eV below the conduction band edge [32,33]. This bulk trap model significantly influenced gate leakage and off-state current in single-layer Al$_2$O$_3$ devices. To evaluate the effect of interface traps on device performance, we extracted the interface trap density (N$_{it}$) from the subthreshold slope (SS) of the simulated transfer characteristics using the following relation as

$$N_{it} = \left(\frac{SS}{\ln 10}\frac{q}{KT} - 1\right)\frac{C_{ox}}{q} \tag{16}$$

Where $C_{ox}$ is the gate oxide per unit area, and SS is extracted from the logarithmic plot of drain current versus gate voltage. These extracted $N_{it}$ values were then incorporated back into the simulation using Silvaco's INTERFACE module. This modeling approach allowed us to assess the quantitative impact of interfacial traps on key parameters such as threshold voltage, mobility, and hysteresis behavior in both single and hybrid dielectric OTFT configurations.

Extracted $N_{it}$ values, summarized in Table 2, reveal a strong dependence on dielectric stack configuration, particularly for the $Al_2O_3$/PVP hybrid system. Notably, an optimized PVP thickness (e.g., 3.6–4.2 nm) substantially suppresses interface trap density, improving carrier mobility and subthreshold performance. It is important to note that the actual interface trap density can vary depending on the fabrication method, material quality, and processing conditions. These factors significantly influence the interfacial properties, such as charge trapping and mobility degradation, and consequently impact the overall device performance.

In order to achieve the calibration between the simulated and fabricated OTFTs, the density of state parameters was adjusted to $W_{TA} = 0.013$, $W_{TD} = 0.12$, $N_{GA} = 1.2 \times 10^{14}$, $N_{GD} = 2 \times 10^{16}$, $E_{GA} = 1.2$, $E_{GD} = 2.9$, and $W_{GA} = 0.05$. The graph shows that there is a slight variation in the calibration results, particularly at the zero voltage of $V_{GS}$. This happened as a result of the reduction in trap densities caused by the hybrid dielectric layer. However, it is important to note that the device slope and the ON current at $V_{DS} = -1.5V$ substantially resemble those of the fabricated device reported by Kraft, Ulrike, et al. [28], as shown in Fig 2(b). Therefore, it may be concluded that the simulation setup is reliable and can be further employed.

## Results and discussion

Selecting the appropriate dielectric material significantly influences the performance of thin-film transistor (TFT) devices. To investigate the interfacial performance of OTFTs, we first introduced an $Al_2O_3$ dielectric layer with a fixed interface trap density of $5 \times 10^{11}$ cm$^{-2}$eV$^{-1}$. This layer was then replaced with a hybrid $Al_2O_3$/PVP dielectric, which has a lower interface trap density, in order to improve the dielectric/semiconductor interface. The hybrid dielectric reduces trap density at the interface, preventing trap formation and minimizing leakage current through the dielectric [36]. This study implements a bilayer dielectric configuration to enhance both the performance and reliability of the device.

### OTFT with single dielectric layer of $Al_2O_3$ and PVP

Fig 3 shows the transfer and output characteristics of a DNTT-based OTFT with single dielectric layers of $Al_2O_3$ and PVP. The electrical parameters were extracted from both the linear and saturation regions of the transfer characteristics,

Table 2. Electrical performance parameters of DNTT- based OTFT with single and hybrid dielectrics.

| Device Characteristics | Single Layer | | Hybrid Layer composition | | |
|---|---|---|---|---|---|
| | PVP | $Al_2O_3$ | P5A95 | P20A80 | P40A60 |
| $V_{TH}$(V) | −0.8 | −0.5 | −0.65 | −0.6 | −0.75 |
| $I_{ON}$ (A) | −1.51x10$^{-5}$ | −8.90x10$^{-5}$ | −1.15x10$^{-4}$ | −1.69x10$^{-4}$ | −1.43x10$^{-4}$ |
| $I_{OFF}$ (A) | −6.19x10$^{-14}$ | −4.39x10$^{-13}$ | −5.17x10$^{-14}$ | −9.24x10$^{-15}$ | −3.50x10$^{-14}$ |
| $I_{ON}$/ $I_{OFF}$ ratio | 2.44x10$^8$ | 2.03x10$^8$ | 2.22x10$^9$ | 1.83x10$^{10}$ | 4.08x10$^9$ |
| SS (mV/dec) | 95 | 100 | 85 | 83 | 84 |
| $G_m$(S) | 1.01x10$^{-5}$ | 5.73x10$^{-5}$ | 6.89x10$^{-5}$ | 1.05x10$^{-4}$ | 8.90x10$^{-5}$ |
| $C_i$[nF/cm$^2$] | 5.6 | 13.7 | 12.78 | 10.64 | 9.56 |
| Mobility sat(cm$^2$/Vs) | 5.16 | 3.86 | 5.28 | 8.43 | 8.17 |
| Contact Resistance (KΩ.cm$^2$) | 63.10 | 30.04 | 5.07 | 4.84 | 6.13 |
| Interface trap density ($N_{it}$) cm$^{-2}$eV$^{-1}$ | 3.42 x 10$^{10}$ | 5.24 x 10$^{11}$ | 2.62 x 10$^{10}$ | 2.39 x 10$^{10}$ | 2.85 x 10$^{10}$ |

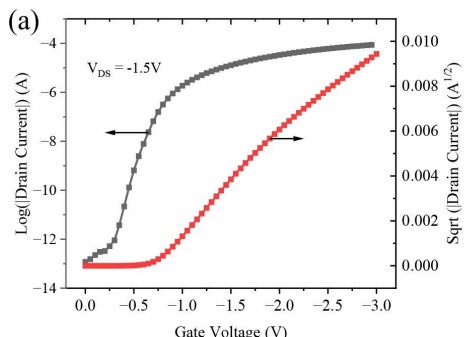
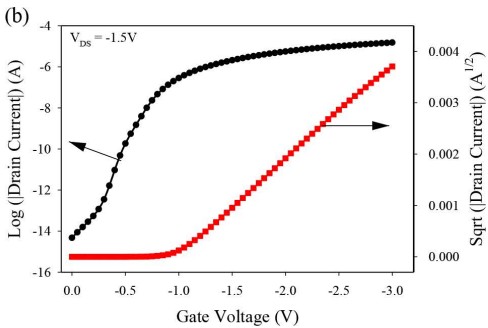

**Fig 3. Transfer characteristics of OTFT with single dielectric of (a) Al$_2$O$_3$ and (b) PVP.**

as extracted from equations (1–6), and summarized them in Table 2. Threshold voltage was determined by plotting the square root of the drain current versus gate voltage, as shown in Fig 3a and 3b.

The OTFT with a single Al$_2$O$_3$ dielectric layer exhibits a lower threshold voltage compared to the PVP-based OTFT, due to its higher dielectric capacitance, which enhances charge accumulation at the semiconductor–dielectric interface, even at low gate voltages.

However, despite this advantage of low threshold voltage, the Al$_2$O$_3$-based OTFT also demonstrates certain limitations, such as lower mobility (3.86 cm$^2$/V·s) and a higher leakage current density of 10$^6$ A/cm$^2$. The mobility was extracted from the linear region of the transfer curve using the standard transconductance method, eq. (6) while the leakage current was obtained under simulated off-state bias conditions ($V_{GS}$=0V, $V_{DS}$=−1.5V). These drawbacks are likely due to the high interface trap density of 5.24 x 10$^{11}$ cm$^{-2}$eV$^{-1}$ at the interface and the inherent characteristics of Al$_2$O$_3$, such as its high density of random dipoles at the semiconductor-dielectric interface.

Although we did not explicitly model spatially distributed dipoles at the Al$_2$O$_3$ interface, their influence was indirectly captured through a higher interface trap density and a Gaussian-distributed bulk trap profile within the Al$_2$O$_3$ layer, centered at 0.6 eV below the conduction band.

This approach allowed the simulation to reflect the leakage-enhancing effects of dipolar disorder through trap-assisted conduction mechanisms, consistent with observed trends and literature reports such as Veres et al [37].

On the other hand, incorporating a PVP, a low-k organic dielectric, results in a smoother interface and low surface energy. This reduction in trap density improves interface quality and mobility due to the inherently low surface energy and smoother morphology of PVP, which minimizes carrier scattering [36]. However, despite these advantages, PVP has a low dielectric constant, which necessitates high operating voltages. Table 2 summarizes the electrical characteristics of the OTFTs with single dielectric layers of Al$_2$O$_3$ and PVP. These findings highlight the trade-offs, demonstrating that while Al$_2$O$_3$ enables low $V_{th}$ operation, it suffers from poor reliability due to traps; PVP, although improving mobility and off-state current, is limited by its low capacitance. These insights underscore the need for a hybrid dielectric approach to balance interface quality and electrical performance in OTFTs.

To address the limitations of single dielectrics, a hybrid bilayer dielectric combining Al$_2$O$_3$ and PVP provides a promising solution for enhancing OTFT performance. By incorporating a thin layer of low-k organic dielectric (PVP) over a high-k inorganic dielectric (Al$_2$O$_3$) the hybrid structure effectively reduces leakage current density, improves the I$_{ON}$/I$_{OFF}$ ratio, and enhances switching performance, thereby reducing power consumption. As shown in Fig 4, the J-V characteristics on a logarithmic scale highlight the advantages of the hybrid dielectric. The Al$_2$O$_3$/PVP devices exhibit a significantly reduced leakage current density of 10$^{-8}$ A/cm$^2$ at -3V, which is two orders of magnitude lower than Al$_2$O$_3$ and one order of magnitude lower than PVP. At $V_{DS}$=−0.05V, thus the bilayer dielectric demonstrates minimal leakage current, enhancing device reliability.

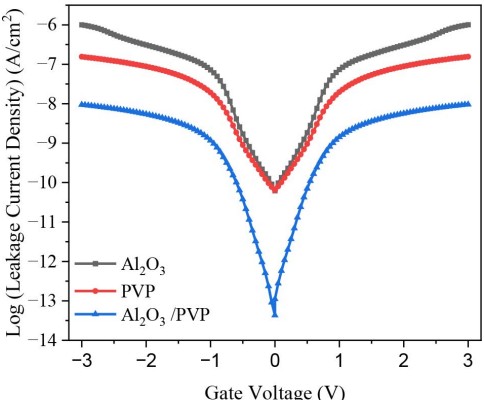

**Fig 4. Shows the leakage current density-voltage (J-V) for single and hybrid dielectric of Al$_2$O$_3$/PVP.**

These results underscore the hybrid dielectric's potential to overcome the trade-offs associated with single dielectrics, offering a balanced approach to improved device performance and stability.

### Effect of hybrid dielectric layer of Al$_2$O$_3$/ PVP

Fig 5 shows the C-V characteristics of MIS capacitors with an Al$_2$O$_3$/PVP hybrid dielectric, using various thickness compositions. The bilayer dielectric's gate capacitance is calculated as per equation (2). The capacitance densities for the compositions P5A95, P20A80, and P40A60 are 12.78 nF/cm² (K = 8.7), 10.64 nF/cm², and 9.56 nF/cm² (K = 6.4), respectively.

As the PVP composition in the hybrid dielectric increases, we observed a gradual decrease in the capacitance due to a reduction in the overall dielectric constant. The lower capacitance results in higher gate voltages being required to accumulate charge carriers at the dielectric interface, which shifts the depletion region toward the positive.

This shift can lead to an increase in the threshold voltage. The dielectric constants of single and hybrid dielectric layers with various compositions of Al$_2$O$_3$ and PVP are evaluated based on capacitance density as per Equation (17) and listed in Table 3.

$$\varepsilon_r = \frac{C_{ox} \cdot t_{die}}{\varepsilon_0}$$

(17)

Where t$_{die}$ $t_{die}$ represents the total thickness of hybrid dielectric layer and C$_{ox}$ represents gate dielectric capacitance.

The transfer and output characteristics of DNTT-based OTFTs with hybrid dielectrics (Al$_2$O$_3$/PVP) were analyzed by varying the composition ratios of Al$_2$O$_3$ to PVP, namely P5A95, P20A80, and P40A60, while maintaining a constant

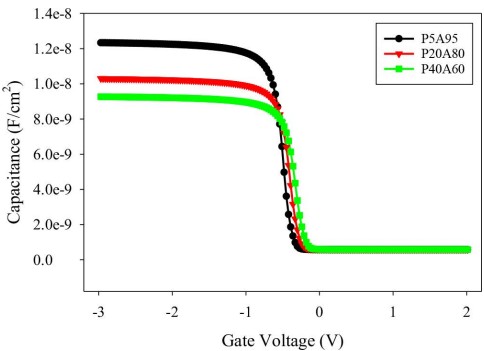

**Fig 5. The C-V characteristics of the various thickness composition of Al$_2$O$_3$/PVP at 1MHz.**

**Table 3. Dielectric properties of the single and bilayer dielectrics.**

| Dielectric materials | Dielectric constant |
| --- | --- |
| Single $Al_2O_3$ | 9.3 |
| Single PVP | 3.8 |
| P5A95 | 8.66 |
| P20A80 | 7.2 |
| P40A60 | 6.4 |

dielectric thickness of 12 nm, as shown in Fig 6. From the output characteristics, it can be observed that all devices exhibit clear linear and saturation regions at a gate voltage of −3 V. The transfer characteristics indicate that increasing the PVP composition in the hybrid dielectric—such as in P5A95 and P20A80 devices—leads to a gradual increase in drain current. This improvement is attributed to enhanced interface quality, despite the reduced gate capacitance. Additionally, the OFF current decreases with higher PVP content, providing better isolation between the electrodes and the conducting channel, which increases the $I_{ON}/I_{OFF}$ ratio (up to $1.83 \times 10^{10}$), making the devices suitable for high-speed applications.

The threshold voltage ($V_{TH}$) was extracted from the plot of $\sqrt{I_{Ds}}$ versus $V_{GS}$, and the subthreshold swing (SS) was obtained from the transfer characteristics of the OTFT at $V_{DS}$ = −1.5Vas shown in Fig 6. The performance parameters, including SS and $V_{TH}$, are critical for improving fast switching and lowering the operating voltage of the device. As the PVP composition in the hybrid dielectric increases, the values of SS and $V_{TH}$ were found to be 85 mV/dec and 0.65 V for P5A95, 83 mV/dec and 0.6 V for P20A80, and 84 mV/dec and 0.75 V for P40A60. It was observed that further increasing the PVP thickness results in a higher threshold voltage and lower drain current due to reduced capacitance. A thinner PVP composition in the hybrid dielectric minimizes carrier scattering and reduces interface trap density, enhancing mobility (up to 8.43 cm²/Vs).

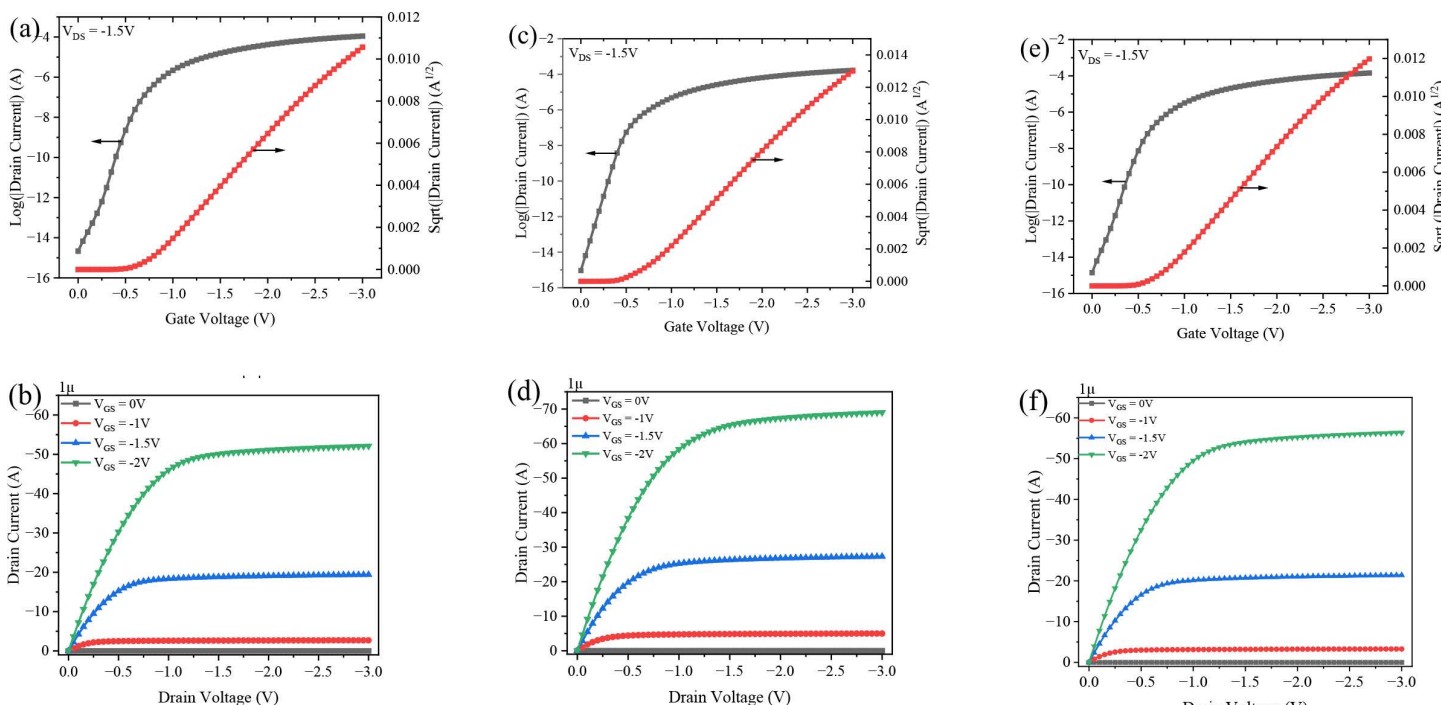

**Fig 6. Transfer and output characteristics of OTFT with various composition of (a),(b) for P5A95 (c),(d) for P20A80 (e)(f) for P40A60.**

However, when the PVP composition exceeds 35%, carriers are trapped at the interface, reducing capacitance and mobility (8.17 cm²/V·s). From our results, the optimal PVP composition in the hybrid dielectric OTFT ranges from 20% to 35% (2.4 nm to 4.2 nm) to prevent carrier trapping at the interface and avoid significant reductions in capacitance and increases in operating voltage. The transfer characteristics of the P20A80 hybrid dielectric OTFT show a notable increase in drain current ($1.69 \times 10^{-4}$ A), surpassing the performance of OTFTs with single $Al_2O_3$ or PVP dielectrics by approximately one order of magnitude.

Alghamdi and Noweir Ahmad [38] developed the transition voltage method (TVM) to extract the contact resistance ($R_c$) from the output characteristics of an OTFT when it is biased in the saturation region. The contact resistance was measured at different values of $V_{GS}$ and expressed in Equation (18).

$$R_C = 2\frac{\sqrt{V_{GS} - V_{TH}}\left[\sqrt{2V_{tr}} - \sqrt{V_{GS} - V_{TH}}\right]}{I_{DSAT}}$$

(18)

$I_{DSAT}$ represents the saturation current, while $V_{tr}$ represents the voltage at which the OTFT transitions from the linear to the saturation region. Therefore, the TVM method is applicable to all devices that exhibit a transition from linear to saturation. Fig 7. shows how the transition voltage method (TVM) is used to extract the contact resistance from the output characteristics of OTFTs with single and hybrid dielectrics. It was observed that hybrid dielectrics exhibit a lower transition voltage compared to single dielectrics, meaning that the transition from the linear to saturation region occurs at lower drain voltages. The contact resistance is significantly reduced to 4.8 KΩ.cm² in OTFTs with an $Al_2O_3$/PVP hybrid dielectric. This reduction is attributed to fewer traps and defects at the interface, which decreases the Schottky barrier height and allows charge carriers to be injected more easily into the channel from the electrodes.

Fig 8(a) illustrates the impact of contact resistance on the drain current in OTFTs with both single and hybrid dielectrics, demonstrating how reduced contact resistance enhances current flow. Fig 8(b). compares the extracted contact resistance ($R_c$.W) of top-contact OTFTs with single and hybrid dielectrics of $Al_2O_3$ and PVP. The contact resistance decreases gradually with increasing gate voltage, which is consistent with previous studies [39,40]. Notably, the hybrid dielectric of $Al_2O_3$/PVP reduces $R_c$ more effectively than single dielectric layers, with Rc extracted at $V_{GS} = -2$ V decreasing from 63.5 KΩ.cm² to 4.8 KΩ.cm². This reduction in contact resistance improves the overall functionality of the device. In this work, we observe that the decrease in contact resistance is primarily due to the reduction in carrier dispersion, improved interface quality, lower trap density, and enhanced charge carrier injection from the electrodes.

Optimizing the PVP composition and thickness in the $Al_2O_3$/PVP hybrid dielectric of OTFTs is a promising approach to improving device performance and reducing contact resistance. Fig 9(a) shows the $I_{ON}$ and $I_{OFF}$ values for OTFTs with

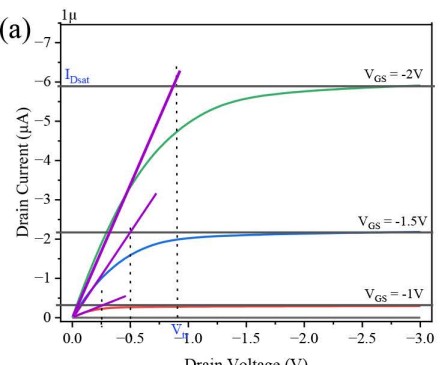 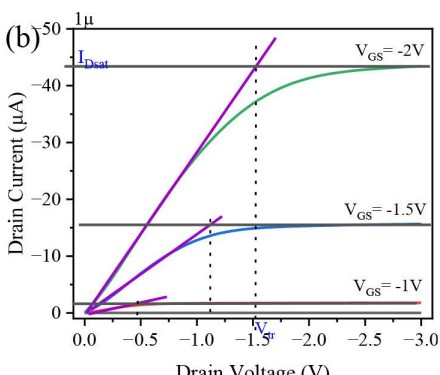 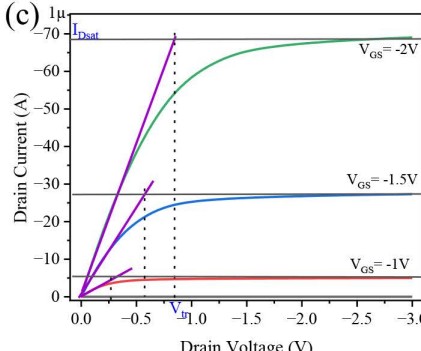

**Fig 7. The contact resistance has been extracted from the output characteristics of OTFT (a) PVP (b) $Al_2O_3$ (c) $Al_2O_3$/PVP.**

single and hybrid dielectrics. The $I_{ON}$ increases from $-1.51 \times 10^{-5}$ A to $-1.69 \times 10^{-4}$ A with hybrid dielectrics, compared to single dielectric layers, due to reduced contact resistance and improved interface quality. Additionally, the $I_{OFF}$ is reduced in hybrid dielectric OTFTs, leading to an $I_{ON}/I_{OFF}$ ratio of $1.83 \times 10^{10}$, which is suitable for high-speed applications. The transconductance ($g_m$) is a key parameter for evaluating device linearity, power efficiency, and amplification capabilities in OTFTs. As shown in Fig 9(b), in single dielectrics ($Al_2O_3$ and PVP), transconductance increases with gate bias until a peak is reached, after which it degrades due to increased charge scattering at higher voltages. This effect is minimized by employing a hybrid dielectric layer ($Al_2O_3$/PVP), which reduces interface trap states at the dielectric-semiconductor interface, resulting in better device stability and performance.

**Effect of interfacial properties of single and hybrid dielectrics of $Al_2O_3$/PVP.** The efficiency of OTFTs is significantly influenced by the properties of the gate dielectric, particularly at the dielectric-semiconductor interface, which directly affects the injection and transport of charge carriers. To enhance the overall performance and stability of the device, we adopted a bilayer dielectric structure composed of a high-k inorganic layer ($Al_2O_3$) and an organic interfacial layer (PVP). This hybrid configuration exploits the advantages of both materials: the high dielectric constant of $Al_2O_3$ ($\kappa \approx 9.3$) increases gate capacitance and promotes low-voltage operation, while the polymeric PVP layer offers a smooth

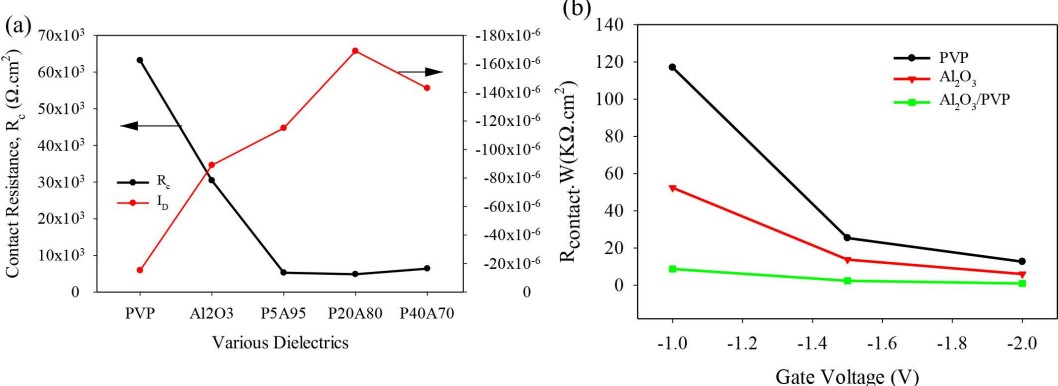

**Fig 8. (a) Impact of contact resistance on drain current (b) Rc.W at different $V_{GS}$ for single and hybrid dielectrics of OTFT.**

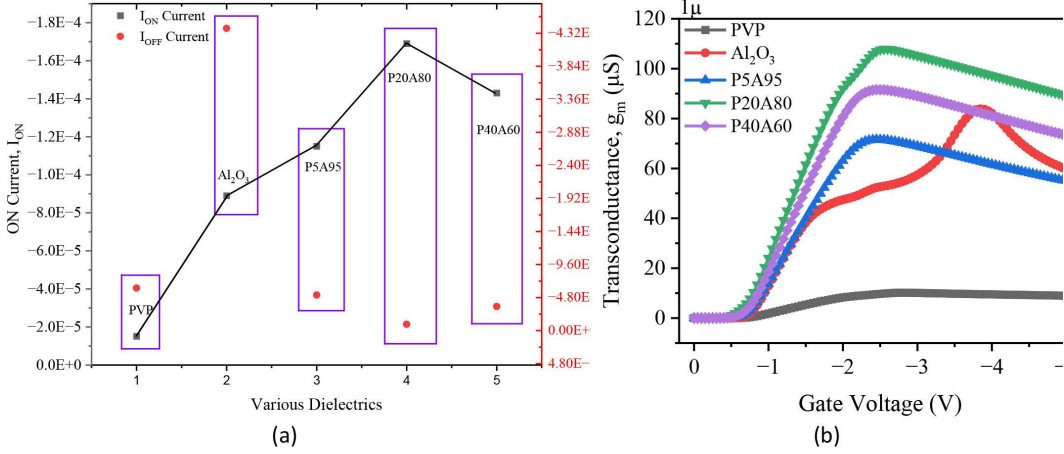

**Fig 9. (a) $I_{ON}$ versus $I_{OFF}$ (b) transconductance of single and various hybrid dielectric compositions.**

surface morphology and a lower density of interface traps that otherwise impede carrier mobility and degrade contact behavior.

The gate dielectric stack has a critically affects energy-level alignment and charge injection at the source/drain contact interface. In single $Al_2O_3$ dielectric OTFT as shown in Fig 10(a), direct contact between $Al_2O_3$ and the DNTT semiconductor often introduces a significant density of interface traps, arising from lattice mismatch, surface roughness, and hydroxyl-related dipoles at the oxide surface. These traps serve as scattering centers and contribute to Fermi level pinning, which inhibits efficient carrier accumulation and widens the Schottky barrier at the metal–semiconductor junction, thereby increasing contact resistance.

In contrast, the hybrid $Al_2O_3$/PVP dielectric stack as shown in Fig 10(b), introduces a thin PVP layer at the dielectric-semiconductor interface. Due to its organic nature and surface energy compatibility with DNTT, PVP minimizes interfacial trap density and promotes improved molecular ordering at the interface. As a result, the trap-induced Fermi level pin-ning at the source/drain contact edges is reduced, enabling more efficient carrier injection. Consequently, the effective Schottky barrier width is reduced, facilitating thermionic emission or tunneling-based carrier injection and lowering contact resistance. Furthermore, the reduced interfacial trap density allows a greater portion of the gate potential to modulate the channel, thereby improving the subthreshold swing and effective carrier mobility.

This interfacial mechanism is illustrated in Fig 10. In Fig 10(a), the single $Al_2O_3$ introduces more interface traps, distort-ing the band profile and impeding carrier injection. In contrast, Fig 10(b) shows that the hybrid dielectric enables smoother band alignment and fewer trap states, enhancing charge injection and lowering contact resistance. Thus, the gate stack not only governs electrostatic control over the channel but also critically influences the contact injection process [41,42].

To further understand the relationship between interfacial properties and device performance, we evaluated leakage current density, C-V characteristics, and hysteresis loss in OTFTs with interface trap densities of $5.24 \times 10^{10}$ cm$^{-2}$eV$^{-1}$, $3.42 \times 10^{10}$ cm$^{-2}$ eV$^{-1}$, and $2.39 \times 10^{10}$ cm$^{-2}$ eV$^{-1}$ for single dielectrics ($Al_2O_3$ and PVP) and the hybrid dielectric ($Al_2O_3$/PVP), respectively. The C-V characteristics in Fig 11(a), demonstrate that the single $Al_2O_3$ dielectric exhibits a negative flat band voltage ($V_{FB}$) of −0.5V, primarily due to the presence of high density of interface traps at the dielectric-semiconductor interface. In contrast, the organic dielectric PVP provides a smoother surface interface, resulting in a more positive flat band voltage of 0.4V. PVP's lower capacitance is attributed to its inherently low dielectric constant. By employing hybrid dielectrics ($Al_2O_3$/PVP), the properties of both organic and inorganic dielectrics are combined, leading to intermediate val-ues of flat band voltage and capacitance. Specifically, the flat-band voltage shifts to −0.1 V, indicating a reduced density of interface traps and an improved interface quality. Additionally, the capacitance is improved to 10 nF, enhancing overall

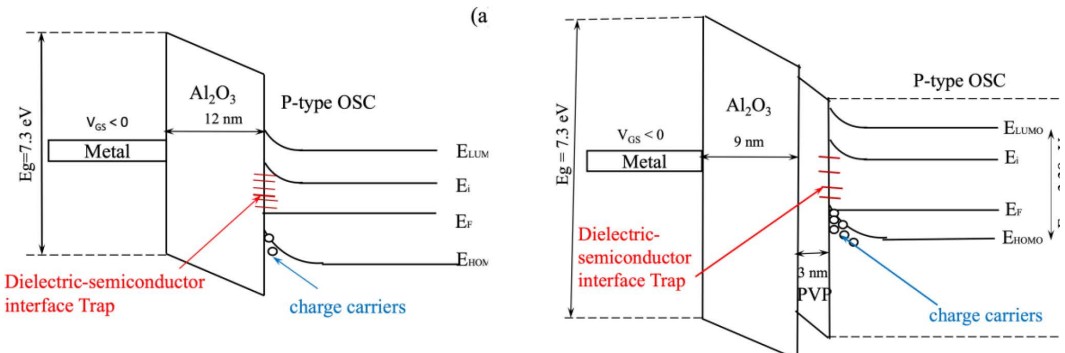

**Fig 10. Schematic band energy diagram of the dielectric-semiconductor interface in a DNTT based OTFT with (a) single $Al_2O_3$ dielectric (b) $Al_2O_3$/PVP gate hybrid dielectric.**

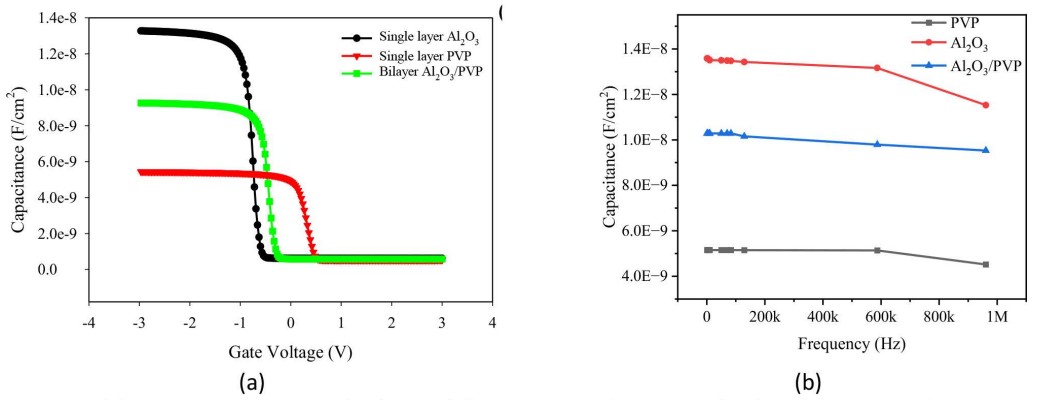

**Fig 11. (a) capacitance-voltage (CV), and (b) capacitance-frequency (C-F) characteristics for single and hybrid dielectric of Al$_2$O$_3$/PVP.**

device performance. Thus, the hybrid dielectric OTFT can operate at lower voltages, making it suitable for applications in wearable electronics and flexible electronics.

Furthermore, as seen in Fig 11(b), the dielectric stability can be investigated by examining the C-F characteristics of various dielectrics of both single Al$_2$O$_3$ and PVP, as well as hybrid dielectric layers of Al$_2$O$_3$ and PVP. As observed, the hybrid dielectric OTFT has demonstrated that the capacitance remains constant over the 1 MHz frequency in comparison to single dielectric layers. This stable capacitance over a wide range of frequencies can ensure the reliability and robustness of the hybrid dielectric layer, contributing to the long-term performance of the device.

Fig 12(a) and 12(b) illustrate the electric field distribution across the single Al$_2$O$_3$ layer and the hybrid Al$_2$O$_3$/PVP layer. The results indicate that the hybrid dielectric improves interface quality, leading to the development of a higher electric

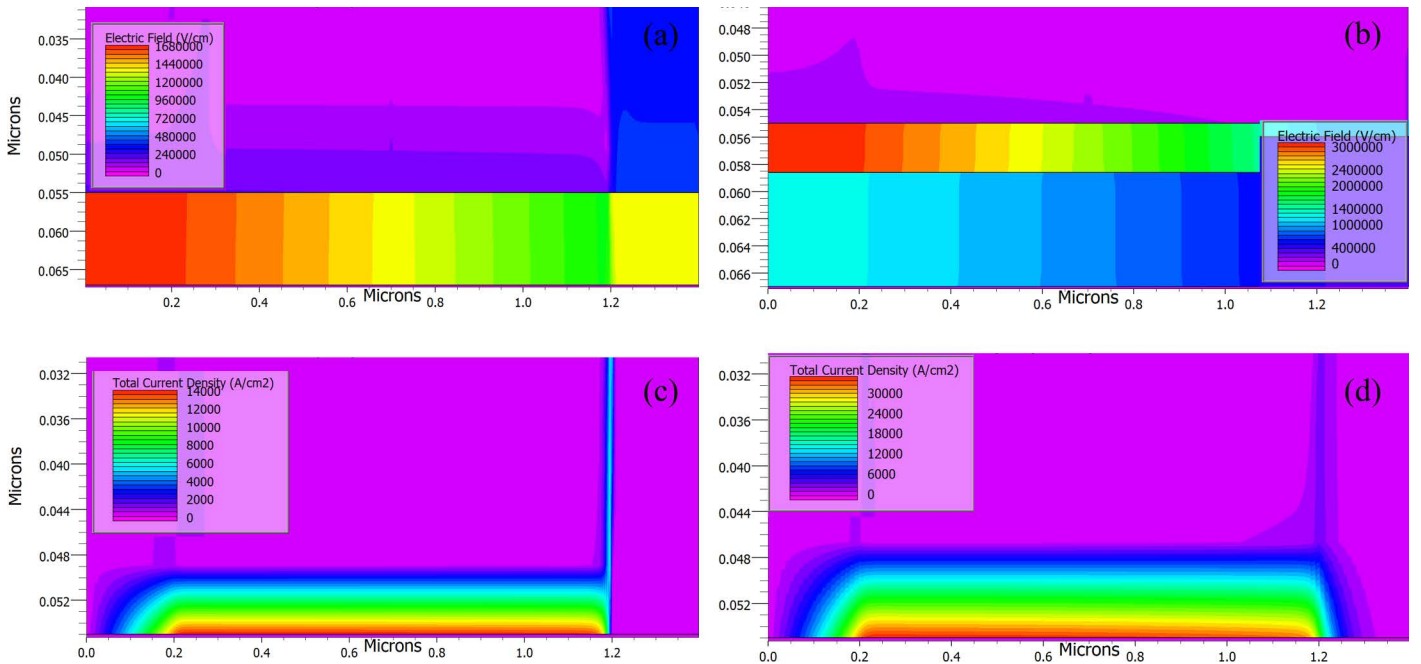

**Fig 12. The contour plots of electric fields of (a) single Al$_2$O$_3$ and (b) hybrid Al$_2$O$_3$/ PVP and total current density of (c) single Al$_2$O$_3$ and (d) hybrid Al$_2$O$_3$ at V$_{DS}$ = −1.5V.**

field at the interface. This stronger electric field facilitates better carrier injection and reduces carrier scattering, resulting in a higher current density across the channel compared to the single $Al_2O_3$ dielectric. Figs 12(c) and 12(d) show the total current density across the conducting channels of the single $Al_2O_3$ and hybrid $Al_2O_3$/PVP dielectrics. These visual representations provide a comprehensive understanding of the performance parameters of the devices.

To further investigate hysteresis loss in OTFTs with single and hybrid dielectrics, we measured the C-V curves in two opposite gate voltage sweep directions, as shown in Fig 13. The gate voltage sweeps ranged from −3 V to +3 V and back to −3 V. In the forward sweep, the C-V curves for both dielectrics shift towards the negative voltage direction, primarily due to hole trapping. During the backward sweep, the C–V curves for the single-dielectric $Al_2O_3$ device exhibit a notable shift toward the positive voltage direction. This is attributed to strong charge trapping at the semiconductor–dielectric interface, which leads to a significant hysteresis window of approximately 0.64 V. Specifically, we employed the INTERFACE and INTTRAP models to simulate the dynamic trapping and de-trapping of charge carriers during gate voltage sweeps. These models allow for energy-distributed traps with finite capture and emission times, enabling the simulation to account for asymmetrical trap kinetics during forward and backward voltage sweeps. This approach accurately reproduces the observed hysteresis in the $Al_2O_3$-based device.

In contrast, the hybrid dielectric shows minimal shift in the C-V curves, as indicated in Fig 13(b). The backward sweep exhibits negligible voltage shift, resulting in a hysteresis window as small as 0.02 V. This reduction is attributed to the improved quality of the PVP/DNTT organic-organic interface, which limits trap formation and suppresses charge accumulation at the dielectric boundary. This reduction in hysteresis underscores the effectiveness of the hybrid dielectric in facilitating charge carrier movement from the electrodes to the active semiconductor, largely due to the lower contact resistance. Overall, these results underscore the critical role of dielectric interface engineering in minimizing hysteresis loss and enhancing stability in OTFTs.

The short-channel OTFT, utilizing a hybrid dielectric of $Al_2O_3$ and polyvinyl phenol (PVP), exhibits improved performance over other hybrid dielectrics by effectively controlling contact resistance and optimizing dielectric dimensions. A summary of these comparisons is provided in Table 4. The short channel (1 μm) OTFTs offers significant advantages for next-generation flexible and high-performance electronics by leveraging the high dielectric constant of $Al_2O_3$ and the organic interface properties of PVP. This combination results in superior interface quality, leading to a steeper subthreshold slope of 83 mV/dec, a higher $I_{ON}/I_{OFF}$ ratio, and a lower threshold voltage, making it suitable for low-power applications compared to other dielectrics.

These improvements make the OTFT particularly well-suited for use in biosensors, where high sensitivity and stability are crucial for detecting biological molecules, as well as in high-speed electronics, where it can operate effectively at higher frequencies. This innovation not only enhances current applications but also opens possibilities for future

(a)

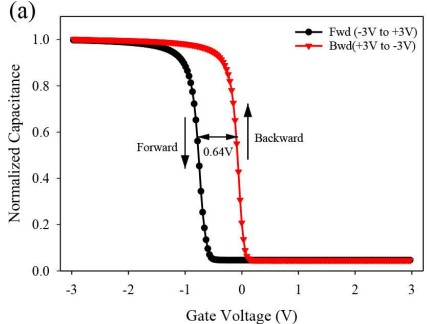

(b)

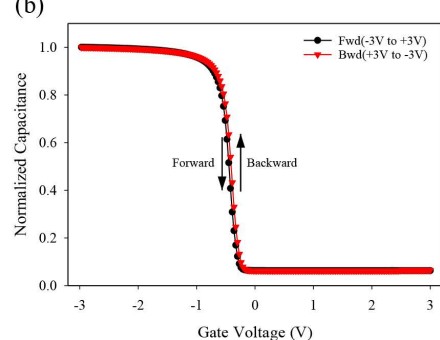

**Fig 13. C-V hysteresis curves of (a)single $Al_2O_3$ (b) $Al_2O_3$/PVP as hybrid dielectrics for OTFT at 1MHz.**

 

**Table 4. Comparison of electrical performances organic TFTs based on various channel lengths and hybrid dielectrics.**

| Refrences | Hybrid dielectric | Organic Semiconductor | Channel Length (μm) | $I_{ON}/I_{OFF}$ Ratio | SS(V/dec) | Threshold Voltage(V) | Operating Voltage (V) |
|---|---|---|---|---|---|---|---|
| [43] | PMMA/$Ta_2O_5$ | Pentacene | 100 | $4.3 \times 10^4$ | 1.04 | −3.5 | 0 to -20V |
| [44] | PVP/$Al_2O_3$ | Pentacene | 90 | $5 \times 10^5$ | 0.5 | −5 | 0 to -10V |
| [21] | PMMA/$ZrO_2$ | Copper ph-thalocyanine (CuPc) | 60 | $1.2 \times 10^3$ | 2.1 | −0.8 | 0 to -40V |
| [45] | KPSPI-6F/PVP | Pentacene | 50 | $1.7 \times 10^5$ | 0.2 | −3.4 | 0 to −5 V |
| This work | PVP/$Al_2O_3$ | DNTT | 1 | $1.83 \times 10^{10}$ | 0.083 | −0.6 | 0 to -3V |

developments in wearable electronics and flexible displays. While the $Al_2O_3$–PVP interface was assumed ideal in this study, moderate interfacial defect densities could influence electric field distribution. Future simulations should examine their impact to further validate the reliability of this hybrid configuration.

## Conclusion

In this paper, we simulated and analyzed a top-contact, bottom-gate OTFT incorporating single and hybrid dielectric configurations, specifically $Al_2O_3$ and PVP. We examined how the interfacial properties of single and hybrid dielectric layers affect the OTFT's performance. Optimizing the PVP thickness in the hybrid layer was found to enhance device performance by reducing contact resistance ($R_c$) in short-channel OTFTs. The optimal PVP thickness was identified as 2.4 nm to 4.2 nm; beyond this range, further increases in PVP thickness led to reduced capacitance and higher operating voltage. Additionally, $R_c$ was shown to significantly impact drain current and charge carrier mobility. The bilayer dielectric also mitigated the hysteresis loss commonly observed with single $Al_2O_3$ dielectrics. Our findings offer a straightforward approach to minimizing short-channel effects, making these devices well-suited for low-voltage, flexible electronics applications such as biosensors and disposable electronics like smart packaging.

## Author contributions

**Conceptualization:** Talla Srinivasa Rao, Matta Durga Prakash.

**Data curation:** Lingala Prasanthi, Subhashini Tata, Rohith Bala Jaswanth B, Matta Durga Prakash, Shovan Kumar Kundu.

**Formal analysis:** Asisa Kumar Panigrahy, Subhashini Tata, Rohith Bala Jaswanth B, Talla Srinivasa Rao, Matta Durga Prakash, Shovan Kumar Kundu.

**Funding acquisition:** Asisa Kumar Panigrahy, Matta Durga Prakash, Shovan Kumar Kundu.

**Investigation:** Lingala Prasanthi, Rohith Bala Jaswanth B, Matta Durga Prakash, Shovan Kumar Kundu.

**Methodology:** Lingala Prasanthi, Subhashini Tata, Talla Srinivasa Rao, Matta Durga Prakash.

**Project administration:** Matta Durga Prakash.

**Resources:** Rohith Bala Jaswanth B, Matta Durga Prakash.

**Software:** Lingala Prasanthi, Matta Durga Prakash.

**Supervision:** Matta Durga Prakash.

**Validation:** Lingala Prasanthi, Matta Durga Prakash, Shovan Kumar Kundu.

**Visualization:** Asisa Kumar Panigrahy, Subhashini Tata, Talla Srinivasa Rao, Matta Durga Prakash, Shovan Kumar Kundu.

**Writing – original draft:** Lingala Prasanthi, Matta Durga Prakash, Shovan Kumar Kundu.

**Writing – review & editing:** Matta Durga Prakash, Shovan Kumar Kundu.

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
