## [Decision Letter · Decision Letter 0]

29 Apr 2025

PONE-D-24-59959Investigation of Gate Dielectric Interface on Contact Resistance of Short Channel Organic Thin Film Transistors (OTFT)PLOS ONE

Dear Dr. Matta,

Thank you for submitting your manuscript to PLOS ONE. After careful consideration, we feel that it has merit but does not fully meet PLOS ONE’s publication criteria as it currently stands. Therefore, we invite you to submit a revised version of the manuscript that addresses the points raised during the review process.

We look forward to receiving your revised manuscript.

Kind regards,

Zeheng Wang

Academic Editor

PLOS ONE

**Journal Requirements:**

Please ensure that your manuscript meets PLOS ONE's style requirements, including those for file naming. The PLOS ONE style templates can be found at https://journals.plos.org/plosone/s/file?id=wjVg/PLOSOne_formatting_sample_main_body.pdf and https://journals.plos.org/plosone/s/file?id=ba62/PLOSOne_formatting_sample_title_authors_affiliations.pdf 2. Please note that PLOS ONE has specific guidelines on code sharing for submissions in which author-generated code underpins the findings in the manuscript. In these cases, we expect all author-generated code to be made available without restrictions upon publication of the work. Please review our guidelines at https://journals.plos.org/plosone/s/materials-and-software-sharing#loc-sharing-code and ensure that your code is shared in a way that follows best practice and facilitates reproducibility and reuse. 3. We note that the grant information you provided in the ‘Funding Information’ and ‘Financial Disclosure’ sections do not match.  When you resubmit, please ensure that you provide the correct grant numbers for the awards you received for your study in the ‘Funding Information’ section.

**Additional Editor Comments:**

The reviewers suggest that the manuscript merits publication in Plos One pending a satisfactory revision addressing all comments from the reviewers.

Reviewers' comments:

Reviewer's Responses to Questions

**Comments to the Author**

1. Is the manuscript technically sound, and do the data support the conclusions?

Reviewer #1: Yes

Reviewer #2: Yes

2. Has the statistical analysis been performed appropriately and rigorously? 

Reviewer #1: Yes

Reviewer #2: N/A

3. Have the authors made all data underlying the findings in their manuscript fully available?

Reviewer #1: Yes

Reviewer #2: Yes

4. Is the manuscript presented in an intelligible fashion and written in standard English?

Reviewer #1: Yes

Reviewer #2: Yes

5. Review Comments to the Author

**Reviewer #1: ** The manuscript presents a valuable numerical study on optimizing hybrid dielectric layers (Al₂O₃/PVP) to address contact resistance and leakage current in short-channel OTFTs. This work is meaningfull in the trade offs between high-κ and low-κ materials to achieve better performance and reliability of OTFTs. However, several questions should be clarified before publication.

1. All the analysis are based on the simulation, while the simulation parameters are not persented,especially for interface trap density of different thickness, which is critical for reliability assessment. More detailed analysis and related reference shoulud be provide. What's more, the bulk trap in Al₂O₃ is also a great concern. The analysis should be provided.

2. Why hybrid dielectric layers is selected as Al₂O₃/PVP, rather than other stacks, such as HfO2/PMMA, should be clarified.

3. This article is primarily about simulation, but there is very little detailed explanation of the specific microscopic mechanisms, especially how the gate stack affects the contact mechanism, which needs to be explained in detail.

4. The simulation assumes a "trap-free interface" at the Al₂O₃/PVP boundary. This simplification may overlook real-world interfacial defects, which are critical for OTFT reliability.

5. It seems that only a very thin layer of low-k dielectric is needed to improve the interface characteristics between the high-k dielectric and OTFT. Are there any relevant methods in actual processes to control the thickness of the low-k dielectric?

**Reviewer #2:**  The manuscript makes a proposal of optimizing and hybrid Al203/PVP dielectric layer of a DNTT organic semiconductor by using a 2D ATLAS Silvaco software.

The strong point of the manuscript is that if hybrid dielectric layer its possible to increase the mobility and reduce the leakage current and use relative low voltage compared to other organic thin film transistors.

The manuscript has some minor typos and it requires a couple of modifications for improving its overall quality.

1.- The manuscript presents a model of 2d transistors, where the appropriate set of differential equations are solved, my question here is how does the simulation deals with the boundary conditions.

2.- At some point it establish that they does not consider defects at the interface of the semiconductor-dielectric interface, however some of the parameters like the hysteresis curve for example clearly depends of the density of these defects.

3.- The contact resistance in given in Ohms instead of Ohm-cm^2, the drain to source is also given the currents instead of a density of currents as i expected from a 2d model.

3.- Figure 2 and 12 the numbers are so tiny that its difficult to read.

4.- There are several parameters just before the results that are not clearly explained what they are.

5.- The model does not consider defects on the interface Al2O3-PVP, could you please provide until what density of defects is still more convenient to use the hybrid layer instead of a single one.

6.- On the results and discussion firs paragraph the density of trap defect has a typo should be order of 10^11

7.- In the manuscript argument that the leakage current and reduce mobility is affected by a random distribution of dipoles that disorganize the energy landscape that make sense but explain how this is included on your model.

There are a couple more comments on the pdf attached.

6. PLOS authors have the option to publish the peer review history of their article (what does this mean? ). If published, this will include your full peer review and any attached files.

**Do you want your identity to be public for this peer review?** For information about this choice, including consent withdrawal, please see our Privacy Policy .

Reviewer #1: **Yes: ** Zirui Wang

Reviewer #2: No

---

## [Author Response · Author response to Decision Letter 1]

23 May 2025

Original Manuscript Number: PONE-D-24-59959

Original Article Title: “Investigation of Gate Dielectric Interface on Contact Resistance of Short Channel Organic Thin Film Transistors (OTFT)”

To: PLOS ONE Editor

Re: Response to reviewers

Dear Editor,

Thank you for allowing a resubmission of revised manuscript and giving us an opportunity to address the reviewers’ comments.

We are uploading (a) our point-by-point response to the comments (below) (response to reviewers, under “Author’s Response Files”), (b) an updated manuscript with yellow highlighting indicating changes (as “Revised Manuscript with Track Changes”), and (c) a clean updated manuscript without highlights (“Manuscript”).

Best regards,

Dr. M. Durga Prakash

Reviewer#1, Concern # 1: All the analysis are based on the simulation, while the simulation parameters are not presented, especially for interface trap density of different thickness, which is critical for reliability assessment. More detailed analysis and related reference should be provide. What's more, the bulk trap in Al₂O₃ is also a great concern. The analysis should be provided.

Author response: We appreciate the reviewer’s emphasis on the importance of clearly presenting simulation parameters and trap-related analysis, particularly in assessing OTFT reliability.

Author action: we have significantly revised the manuscript to include a comprehensive summary of simulation parameters (Table 1), including material properties, mobility models, and geometric details used in the Silvaco Atlas simulations.

Table 1. Summary of structure dimensions and material parameters used in simulation

Classification Parameters Values /units

Physical Dimensions Channel Length (Lch) 1 µm

Width (W) 200 µm

Thickness of DNTT (Tosc) 25 nm

Thickness of single dielectric Al2O3, PVP (Tdie) Fixed at 12 nm

Thickness of hybrid dielectric Al₂O₃ / PVP (Tdie) Varied for different compositions

Thickness of electrodes(Tsd,Tg) 30 nm

Materials

Organic SC-D NTT Bandgap (eg) 3.38 [34]

Permittivity (ε) 3

Electron Affinity (χe) 1.81

Effective DOS of LUMO (Nc) 1e20 cm-3

Effective DOS of HOMO(Nv) 1e20 cm-3

Doping Concentration 1x1015 cm-3

Al2O3 Dielectric Constant 9.3 [35]

Bulk trap density 1x1014 cm-3

PVP Dielectric Constant 3.8

Work function Gold (Au) 5.0 eV

Aluminium (Al) 4.28 eV

PF Mobility Parameters deltaep.pfmob 1.792e-1 eV

betap.pfmob 7.785e5 eV(cm/V)1/2

Temperature (K) 300K

Regarding interface trap density and about bulk trap states in Al2O3, we have added a detailed explanation in the Simulation Methodology section on how Nit was extracted from the subthreshold slope (SS) using Eq. (16). We further tabulated the extracted Nit values for different dielectric configurations in Table 2 and discussed their correlation with dielectric thickness and performance metrics. These findings demonstrate how hybrid dielectric structures (especially PVP-rich stacks) help suppress interface trap density and improve mobility as follows

Table 1 enumerates the material input parameters, including device dimensions, dielectric properties, and mobility models used in the TCAD simulations for OTFTs with hybrid and single-layer dielectrics.

To account for the effect of trapped charges, Poisson’s equation was modified to incorporate spatial charge contributions from ionized traps:

div(ε∇Ψ)=q(n-p-N_tD^++N_A^-)-Q_T (10)

where Q_T=q(N_tD^+ +N_tA^- ) represents the total trapped charge density

Here, N_tD^+=trap density × F_tD and N_tA^-=trap density ×F_█(tA@)

Here, N_tD^+ and N_tA^- are the ionized donor- and acceptor-like trap densities respectively and F_tD and F_tA are their respective ionization probabilities. Organic semiconductors inherently, contain a tail and deep -level traps within the band gap, which are described using both exponential (tail) and Gaussian (deep) density-of-states (DOS) functions.

The equations that describe these terms are as follows

g(E)=g_TA (E)+g_TD (E)+g_GA (E)+g_GD (E) (11)

where the tail and deep level bands are modeled with an exponential and Gaussian distributions, respectively, as:

g_TA (E)=N_TA exp[( E - E_(c ))/W_TA ] (12)

g_TD (E)=N_TD exp[( E_V - E )/W_TD ] (13)

g_GA (E)=N_GA exp[-[(E_GA-E)/W_GA ]^2 ] (14)

g_GD (E)=N_GD exp[-[(E-E_GD)/W_GD ]^2 ] (15)

Here, E is the trap energy, EC is the conduction band energy, EV is the valence band energy and the subscripts (T, G, A, D) stand for tail, Gaussian (deep level), acceptor and donor states respectively.

The quantities NTA, NTD, NGA, NGD and WTA, WTD, WGA, WGD are the fitting parameters of TRAP models. In our simulations, both interface and bulk trap effects were incorporated using Silvaco Atlas INTDEFECTS and INTERFACE models to accurately capture semiconductor-dielectric and semiconductor-contact interfacial behaviors. Specifically, we implemented a Gaussian distribution of bulk traps in the Al2O3layer with a peak trap density of 1×1014 cm−3, centered at 0.6 eV below the conduction band edge [32][33]. This bulk trap model significantly influenced gate leakage and off-state current in single-layer Al2O3 devices. To quantify interface trap density (Nit), we extracted values from the subthreshold slope (SS) of the transfer characteristics using the established relation.

N_it=(SS/ln10 q/KT-1) C_ox/q (16)

Where Cox is the gate oxide capacitance per unit area, and SS is extracted from the logarithmic plot of drain current versus gate voltage. Extracted Nit values, summarized in Table 2, reveal a strong dependence on dielectric stack configuration, particularly for the Al2O3/PVP hybrid system. Notably, an optimized PVP thickness (e.g., 3.6–4.2 nm) substantially suppresses interface trap density, improving carrier mobility and subthreshold performance. It is important to note that Nit values are sensitive to process-dependent factors, including surface roughness, polymer cross-linking, and metal-oxide interface quality.

[32] Shi, Yuanyuan, et al. "Investigation of bulk traps by conductance method in the deep depletion region of the Al 2 O 3/GaN MOS device." Nanoscale research letters 12 (2017): 1-6.

[33] Deng, Kexin, et al. "Insight into the suppression mechanism of bulk traps in Al2O3 gate dielectric and its effect on threshold voltage instability in Al2O3/AlGaN/GaN metal-oxide-semiconductor high electron mobility transistors." Applied Surface Science 638 (2023): 158000.

These revisions ensure that both interface and bulk defect contributions are clearly modeled, quantified, and linked to the device's electrical performance. We thank the reviewer for prompting us to clarify and expand this critical aspect of our simulation framework.

Reviewer#1, Concern # 2: Why hybrid dielectric layers is selected as Al2O3/PVP, rather than other stacks, such as HfO2/PMMA, should be clarified.

Author response: We appreciate the reviewer’s suggestion to clarify the rationale for selecting the Al2O3/PVP hybrid dielectric system. Our choice was guided by several considerations:

Dielectric Constant Complementarity: Al2O3 offers a relatively high dielectric constant (~9.3), which allows for low-voltage operation, while PVP has a low-κ (~4.7) value, which is effective in suppressing gate leakage and minimizing interface traps.

Interface Compatibility with DNTT: PVP forms a smooth organic–organic interface with the DNTT semiconductor, facilitating molecular ordering and reducing charge scattering. This is particularly advantageous compared to PMMA, which is less favourable in forming high-quality interfaces with DNTT.

Thermal and Mechanical Stability: PVP films can be cross-linked thermally, offering stable and uniform thin layers. This enables the realization of sub-5 nm interfacial layers critical for hybrid stack tuning.

Literature Precedence and Availability: Al2O3 and PVP are both widely used and well-characterized in OTFT research. Prior studies (e.g., Kraft et al. [27]) have demonstrated the compatibility of this stack with short-channel, high-mobility OTFTs.

While HfO2/PMMA is a viable alternative, our selection of Al2O3/PVP was based on achieving a balance between process feasibility, device performance, and interface quality specific to our DNTT-based architecture. We have now clarified this material rationale in the revised manuscript.

er.

Author action: We have updated the manuscript to include the importance of Al2O3/PVP hybrid dielectric stack in the introduction section as follows.

In this context, we chose Al2O3/PVP as the hybrid dielectric stack, due to their complementary material properties and proven compatibility with organic semiconductors. Al2O3 offers a high dielectric constant (~9), excellent thermal stability, and favorable electrical performance at low voltages. Meanwhile, PVP contributes hydrophobicity, low interface trap density, mechanical flexibility, and biocompatibility. This specific combination was selected over other stacks because (i) Al2O3 provides better interfacial quality and lower trap density compared to high -K dielectric materials which often introduces instability in OTFTs, and (ii) PVP exhibits better uniform film formation and fewer residual polar groups than organic dielectrics, which improves the semiconductor–dielectric interface. These advantages make Al2O3/PVP a robust choice for hybrid dielectric engineering in high performance OTFTs.

Reviewer#1, Concern # 3: This article is primarily about simulation, but there is very little detailed explanation of the specific microscopic mechanisms, especially how the gate stack affects the contact mechanism, which needs to be explained in detail.

Author response: We sincerely thank the reviewer for highlighting the need for a more detailed microscopic explanation of how the gate dielectric stack impacts the contact resistance mechanism.

Author action: We have significantly expanded the discussion section, to clarify the electrostatic and interfacial mechanisms underlying this effect, supported by literature references and energy band considerations.

The efficiency of OTFTs is significantly influenced by the properties of the gate dielectric, particularly at the dielectric-semiconductor interface, which directly affects the injection and transport of charge carriers. To enhance the overall performance and stability of the device, we adopted a bilayer dielectric structure composed of a high-k inorganic layer (Al2O3) and an organic interfacial layer (PVP). This hybrid configuration exploits the advantages of both materials: the high dielectric constant of Al2O3 (κ ≈ 9.3) increases gate capacitance and promotes low-voltage operation, while the polymeric PVP layer offers a smooth surface morphology and a lower density of interface traps that otherwise impede carrier mobility and degrade contact behavior.

The gate dielectric stack has a critically affects energy-level alignment and charge injection at the source/drain contact interface. In single Al2O3 dielectric OTFT illustrated in Fig 10(a), direct contact between Al2O3 and the DNTT semiconductor often introduces a significant density of interface traps, arising from lattice mismatch, surface roughness, and hydroxyl-related dipoles at the oxide surface. These traps serve as scattering centers and contribute to Fermi level pinning, which inhibits efficient carrier accumulation and widens the Schottky barrier at the metal–semiconductor junction, thereby increasing contact resistance.

In contrast, the hybrid Al2O3 /PVP dielectric stack as shown in Fig 10(b), introduces a thin PVP layer at the dielectric-semiconductor interface. Due to its organic nature and surface energy compatibility with DNTT, PVP minimizes interfacial trap density and promotes improved molecular ordering at the interface. As a result, the trap-induced Fermi level pinning at the source/drain contact edges is reduced, enabling more efficient carrier injection. Consequently, the effective Schottky barrier width is reduced, facilitating thermionic emission or tunneling-based carrier injection and lowering contact resistance. Furthermore, the reduced interfacial trap density allows a greater portion of the gate potential to modulate the channel, thereby improving the subthreshold swing and effective carrier mobility.

This interfacial mechanism is illustrated in Fig 10. In Figs 10a, the single Al2O3 introduces more interface traps, distorting the band profile and impeding carrier injection. In contrast, Fig. 10b shows that the hybrid dielectric enables smoother band alignment and fewer trap states, enhancing charge injection and lowering contact resistance. Thus, the gate stack not only governs electrostatic control over the channel but also critically influences the contact injection process [41][42].

[41] Kim, Kang Dae, and Chung Kun Song. "Low-voltage organic thin-film transistors using a hybrid gate dielectric consisting of aluminum oxide and poly (vinyl phenol)." Japanese journal of applied physics 49.11R (2010): 111603.

[42] Soltani, B., M. Babaeipour, and A. Bahari. "Studying electrical characteristics of Al 2 O 3/PVP nano-hybrid composites as OFET gate dielectric." Journal of Materials Science: Materials in Electronics 28 (2017): 4378-4387

Reviewer#1, Concern # 4 The simulation assumes a "trap-free interface" at the Al2O3 /PVP boundary. This simplification may overlook real world interfacial defects, which are critical for OTFT reliability.

Author response: We thank the reviewer for pointing out this inconsistency. We acknowledge that an earlier version of the manuscript may have caused confusion by suggesting that interface defects were not considered. To clarify:

Our simulation explicitly includes defects at the semiconductor–dielectric interface, particularly at the DNTT/PVP and DNTT/Al₂O₃ boundaries.

Interface trap density (Nit) was extracted from the subthreshold slope using Eq. (16) and implemented using Silvaco’s INTERFACE and INTTRAP models.

These interface traps were essential for accurately simulating hysteresis behavior, subthreshold swing, and leakage current in the OTFTs.

We have revised the manuscript accordingly to ensure consistency and have removed the misleading wording suggesting that the interface was considered trap-free.

Author action: we have updated the manuscript, included interface trap modeling in our simulation framework. Specifically:

To evaluate the effect of interface traps on device performance, we extracted the interface trap density (Nit) from the subthreshold slope (SS) of the simulated transfer characteristics using the following relation

N_it=(SS/ln10 q/KT-1) C_ox/q (16)

Where Cox is the gate oxide per unit area, and SS is extracted from the logarithmic plot of drain current versus gate voltage. These extracted Nit values were then incorporated back into the simulation using Silvaco’s INTERFACE module. This modeling approach allowed us to assess the quantitative impact of interfacial traps on key parameters such as threshold voltage, mobility, and hysteresis behavior in both single and hybrid dielectric OTFT configurations.

Reviewer#1, Concern # 5 It seems that only a very thin layer of low-k dielectric is needed to improve the interface characteristics between the high-k dielectric and OTFT. Are there any relevant methods in actual processes to control the thickness of the low-k dielectric?

Author response: We thank the reviewer for this important observation. Yes, in practical fabrication processes, there are well-established methods to precisely control the thickness of ultra-thin low-k polymer dielectrics such as poly(4-vinylphenol) (PVP)

Author action: We have revised the manuscript by including relevant fabrication techniques in the simulation section, supported by appropriate references.

Although such ultra-thin polymer layers are challenging to fabricate, they are achievable using controlled spin-coating techniques, where the film thickness is modulated through solution concentration and spin speed. Fan et al. [29] demonstrated the succes

---

## [Decision Letter · Decision Letter 1]

8 Jun 2025

Investigation of Gate Dielectric Interface on Contact Resistance of Short Channel Organic Thin Film Transistors (OTFT)

PONE-D-24-59959R1

Dear Dr. Matta,

We’re pleased to inform you that your manuscript has been judged scientifically suitable for publication and will be formally accepted for publication once it meets all outstanding technical requirements.

Kind regards,

Zeheng Wang

Academic Editor

PLOS ONE

Additional Editor Comments (optional):

Reviewers' comments:

Reviewer's Responses to Questions

**Comments to the Author**

1. If the authors have adequately addressed your comments raised in a previous round of review and you feel that this manuscript is now acceptable for publication, you may indicate that here to bypass the “Comments to the Author” section, enter your conflict of interest statement in the “Confidential to Editor” section, and submit your "Accept" recommendation.

Reviewer #1: All comments have been addressed

Reviewer #2: All comments have been addressed

2. Is the manuscript technically sound, and do the data support the conclusions?

Reviewer #1: Yes

Reviewer #2: Yes

3. Has the statistical analysis been performed appropriately and rigorously? 

Reviewer #1: Yes

Reviewer #2: N/A

4. Have the authors made all data underlying the findings in their manuscript fully available?

Reviewer #1: Yes

Reviewer #2: Yes

5. Is the manuscript presented in an intelligible fashion and written in standard English?

Reviewer #1: Yes

Reviewer #2: Yes

6. Review Comments to the Author

Reviewer #1: (No Response)

Reviewer #2: After carefully reviewing the manuscript i found that each and everyone of my concerns are correctly addressed therefore i recommend to accept the manuscript without further delay.

7. PLOS authors have the option to publish the peer review history of their article (what does this mean? ). If published, this will include your full peer review and any attached files.

**Do you want your identity to be public for this peer review?** For information about this choice, including consent withdrawal, please see our Privacy Policy .

Reviewer #1: **Yes: ** Zirui Wang

Reviewer #2: No

---

## [Editor Report · Acceptance letter]

PONE-D-24-59959R1

PLOS ONE

Dear Dr. Prakash,

I'm pleased to inform you that your manuscript has been deemed suitable for publication in PLOS ONE. Congratulations! Your manuscript is now being handed over to our production team.

Kind regards,

on behalf of

Dr. Zeheng Wang

Academic Editor

PLOS ONE